# Tenecteplase or Alteplase Better in Patients with Acute Ischemic Stroke Due to Large Vessel Occlusion: A Single Center Observational Study

**DOI:** 10.3390/medicina58091169

**Published:** 2022-08-28

**Authors:** Agnete Teivane, Kristaps Jurjāns, Jānis Vētra, Jekaterina Grigorjeva, Karlis Kupcs, Rytis Masiliūnas, Evija Miglāne

**Affiliations:** 1Faculty of Residency, Riga Stradiņš University, LV-1007 Riga, Latvia; 2Neurology Department, Pauls Stradiņš Clinical University Hospital, LV-1002 Riga, Latvia; 3Department of Neurology and Neurosurgery, Riga Stradiņš University, LV-1007 Riga, Latvia; 4Department of Neurology, The Red Cross Medical College of Riga Stradiņš University, LV-1007 Riga, Latvia; 5Faculty of Medicine, Riga Stradiņš University, LV-1007 Riga, Latvia; 6Department of Radiology, Riga Stradiņš University, LV-1007 Riga, Latvia; 7Institute of Diagnostic Radiology, Pauls Stradinš Clinical University Hospital, LV-1002 Riga, Latvia; 8Center of Neurology, Vilnius University, LT-08661 Vilnius, Lithuania

**Keywords:** acute ischemic stroke, intravenous thrombolysis, Tenecteplase, large vessel occlusion, spontaneous recanalization

## Abstract

*Background and Objectives*: The study aimed to investigate the efficacy of intravenous thrombolysis with Tenecteplase before thrombectomy for acute ischemic stroke (AIS) patients compared with previous results using Alteplase. Previous trials for Tenecteplase have indicated an increased incidence of vascular reperfusion. In April 2021, we started to primarily give Tenecteplase to patients eligible to undergo thrombectomy. *Materials and Methods*: In this retrospective observational single-center non-randomized study, we analyzed directly admitted patients with AIS who had occlusion of the internal carotid, middle cerebral, or basilar artery and who underwent thrombectomy, as well as the recanalization rate for these patients at the first angiographic assessment (mTICI score 2b–3), and complications. *Results*: We included 184 patients (demographic characteristics did not differ between Tenecteplase and Alteplase groups (mean age 68.4 vs. 73.0 years; female sex 53.3% vs. 51.1%, NIHSS 14 (IQR 4–26) vs. 15 (2–31). Forty-five patients received Tenecteplase and 139 Alteplase before endovascular treatment (EVT). Pre-EVT (endovascular treatment) recanalization was more likely to occur with Tenecteplase rather than Alteplase (22.2% vs. 8.6%, *p* = 0.02). Successful reperfusion (mTICI 2b–3) after EVT was achieved in 155 patients (42 (93.4%) vs. 113 (81.3), *p* = 0.07). Hemorrhagic imbibition occurred in 15 (33.3%) Tenecteplase-treated patients compared with 39 (28.1%) Alteplase-treated patients (*p* = 0.5). Patients treated with Tenecteplase had higher odds of excellent functional outcome than Alteplase-treated patients (Tenecteplase 48.6% vs. Alteplase 26.1%; OR 0.37 (95% CI 0.17–0.81), *p* = 0.01). *Conclusions*: Tenecteplase (25 mg/kg) could have superior clinical efficacy over Alteplase for AIS patients with large-vessel occlusion (LVO), administered before EVT. The improvement in reperfusion rate and the better excellent functional outcome could come without an increased safety concern.

## 1. Introduction

Acute ischemic stroke (AIS) is a severe and life-threatening disease, particularly when caused by a large-vessel occlusion (LVO), which is associated with more severe neurological deficits and worse functional outcomes [1,2]. Intravenous thrombolysis (IVT) with Alteplase has been the only proven thrombolytic agent in patients with AIS for nearly three decades. Only during the last few years more evidence for the role of another thrombolytic agent, Tenecteplase, has been uncovered. 

The very first randomized trial comparing Tenecteplase and Alteplase in computed tomographic (CT) perfusion selected AIS patients, published in 2012, indicated that Tenecteplase could be superior concerning the coprimary endpoints of reperfusion and clinical improvement at 24 h [3]. Even though the subsequent meta-analyses of Tenecteplase trials did not confirm its superiority over Alteplase in non-selected AIS patients, non-inferior safety and efficacy relative to Alteplase and superiority for early recanalization were demonstrated [4,5].

In 2018, the EXTEND-IA TNK trial found that Tenecteplase was associated with increased early recanalization and better 90-day functional outcome over Alteplase among a subgroup of patients with AIS due to LVO treated within 4.5 h from symptom onset before endovascular treatment (EVT) [6]. These findings were later echoed by real-life data [7,8]. Based on the previous trials, the current international IVT guidelines include a recommendation to apply Tenecteplase 0.25 mg/kg over Alteplase before EVT only, but the quality of evidence is low and the strength of the recommendation is weak [9,10]. Therefore, there is a necessity to gather more information about the efficacy of Tenecteplase in patients with LVO, eligible for EVT. 

We aimed to investigate the real-life situation of (1) excellent functional outcome and (2) mortality in AIS patients with LVO, treated with IVT using Tenecteplase versus Alteplase before EVT in our comprehensive stroke center (CSC) in Latvia.

## 2. Materials and Methods

### 2.1. Study Design and Population

In this single-center cross-sectional retrospective observational study, we included AIS patients with LVO that received bridging therapy with IVT and EVT for 2 years from 1 January 2020, to 31 December 2021 in one, CSC in Riga, Latvia (Department of Neurology, Pauls Stradiņš Clinical University Hospital). 

Inclusion criteria: (1) age ≥ 18 years old; (2) clinical diagnoses of AIS with a measurable neurologic deficit in the National Institute of Health Stroke Scale (NIHSS) presenting within the 4.5 h window from symptom onset (NIHSS score ≥ 1); (3) LVO (internal carotid artery, first segment of middle cerebral artery, second segment of middle cerebral artery or basilar artery) [6], confirmed either by CT angiography, magnetic resonance (MR) angiography or transcranial Doppler; and (4) eligibility for IVT according to international recommendations [10,11]; (5) signed informed consent by the patient or his/her proxies for the treatment with Tenecteplase (January 2021–December 2021) or the treatment with Alteplase (January 2020–March 2020). COVID-19 positive patients were not excluded from the study. Patients were divided into two groups depending on the thrombolytic agent received—the Alteplase group and the Tenecteplase group. As Tenecteplase was not available at the time for routine stroke treatment, the Alteplase group consists of patients treated in 2020, whereas the Tenecteplase group includes patients treated in 2021. In total, 184 patients met the inclusion criteria and were assessed during the study period.

### 2.2. Outcome Measures

The patients with AIS at admission were thoroughly evaluated and diagnosed by certified vascular neurologists, who further performed a thrombolysis procedure. The decision for systemic reperfusion with IVT (Alteplase 0.9 mg/kg or Tenecteplase 0.25 mg/kg) was made based on AHA/ASA [10] or European Stroke Organisation (ESO) recommendations [11]. 

Initial CT, CT angiography, and CT perfusion were performed upon admission and cerebral imaging data were assessed retrospectively by experienced certified vascular neurologists and neuroradiologists. Thrombectomies and digital subtraction angiography (DSA) procedures were performed by interventional radiologists. In addition, a second CT scan was performed on all patients 24 h after stroke to evaluate ischemic changes or hemorrhagic transformation. Evaluation of the hemorrhagic transformation was done by a radiologist that was blind to the outcome. 

The following parameters were prospectively collected for all included patients: (1) Demographic characteristics (age, sex), (2) mean hospital stay (days), (3) history of vascular risk factors (diabetes mellitus, hypertension, current smoking, hypercholesterolemia, coronary artery disease, congestive heart failure, and atrial fibrillation), (4) prior history of stroke/transient ischemic attack. Stroke severity was assessed with NIHSS score at admission, 24 h post IVT, and discharge. Safety outcomes included prevalence of symptomatic intracranial hemorrhage (sICH) and mortality; sICH was defined using standard Safe Implementation of Thrombolysis in Stroke (SITS) registry definitions [12]. 

The following outcome data were collected: time metrics (symptom onset, beginning of brain imaging, IVT, groin puncture, spontaneous recanalization rates—noted during the initial DSA scan.), 24-h parenchymal hemorrhage (PH) on imaging, the prevalence of sICH, successful recanalization (modified Treatment in Cerebral Ischemia (mTICI) 2b–3 score) [13]. Functional independence was measured by a phone interview (the patient or his/her proxies were asked standardized questions to evaluate the mRS score) at 90 days using the modified Rankin Scale (mRS) [14] and was defined as an mRS score of 0–2. Stroke severity and functional outcome (mRS) at discharge and 3 months were assessed by a certified vascular neurologist [15]. 

The study was approved by the local Ethics Committees of Riga Stradiņš University (decision number 2/468/2021). All patients or their proxies gave informed verbal consent to participate in the study.

### 2.3. Statistical Analysis

Results were summarized using “Microsoft Excel 16.0” (Redmond, WA, USA) and analyzed using “IBM SPSS Statistics 27 edition” (Armonk, NY, USA) software. All binary variables were presented as percentages (%), while continuous variables were presented with their corresponding mean values and standard deviations (SDs), in cases of normal distributions, or as medians with interquartile ranges (IQRs) in cases of skewed distributions. Categorical variables were compared with the Chi-square test and continuous variables were assessed for normal distribution with the Kolmogorov–Smirnov test. Statistical comparisons between the two groups were performed using an unpaired *t*-test, Mann–Whitney U-test, as appropriate. 

In-hospital mortality and the 90-day mRS score were dichotomized into functional independence (score 0–2) and dependence (score 3/4–5), as well as death (score 6). The distribution of the 3-month mRS scores between patients treated with Alteplase and Tenecteplase was compared using the Cochran–Mantel–Haenszel test and univariable/multivariable ordinal logistic regression (shift analysis). All efficacy and safety outcomes of interest were further assessed in multivariable binary and ordinal multivariable logistic regression models adjusting for the confounders—age, sex, pre- mRS, onset-to-EVT-time, and baseline NIHSS. All tests were two-sided and *p* values < 0.05 were considered significant. 

## 3. Results

A total of 184 patients underwent thrombolysis and subsequent EVT during the study period. Forty-five patients received Tenecteplase and 139 Alteplase before EVT. Patient demographic characteristics did not differ between Tenecteplase and Alteplase groups (mean age 68.4 vs. 73.0 years, *p* = 0.14; female sex 53.3% vs. 51.1%, *p* = 0.79, respectively). Tenecteplase-treated patients were less likely to have a history of congestive heart failure (33.3% vs. 51.1%, *p* = 0.04) and more likely to have previous cerebrovascular events (33.3% vs. 18.0%, *p* = 0.03). The median baseline NIHSS (14 (IQR 4–26) vs. 15 (2–31), *p* = 0.52), pre-stroke mRS (0 (0–1) vs. 0 (0–1), *p* = 0.26), and onset time to thrombectomy (165 vs. 180 min, *p* = 0.32) were similar between the two groups. Pre-EVT recanalization was more likely to occur with Tenecteplase rather than Alteplase (22.2% vs. 8.6%, *p* = 0.02). Successful reperfusion (mTICI 2b–3) after EVT was achieved in 155 patients (42 (93.4%) vs. 113 (81.3), *p* = 0.07). Angioedema occurred in one (2.2%) Tenecteplase-treated patient compared with zero (0%) Alteplase-treated patients (*p* = 0.08). Hemorrhagic imbibition occurred in 15 (33.3%) Tenecteplase-treated patients compared with 39 (28.1%) Alteplase-treated patients (*p* = 0.5). The European Cooperative Acute Stroke Study (ECASS) hemorrhagic transformation score was evaluated; in the Tenecteplase-treated patient group, the most common score was HI1 (5 (35.7%)), and in the Alteplase-treated patient group, the most common score was PH1 (13 (33.3%)). Subarachnoid hemorrhage occurred in two (4.4%) Tenecteplase-treated patients compared with nine (6.5%) Alteplase-treated patients (*p* = 0.62). All baseline characteristic data previously described in this paragraph can be seen in Table 1.

Patients treated with Tenecteplase had higher odds of excellent functional outcome mRS (0–2) than Alteplase-treated patients upon discharge (Tenecteplase 40% vs. Alteplase 22.3%; OR 0.43 (95% CI 0.21–0.88)), and this difference was statistically significant (*p* = 0.02). Patients treated with Alteplase had higher odds of mRS 3 upon discharge than the Tenecteplase group (Tenecteplase 8.9% vs. Alteplase 20.1%; OR 2.59 (95% CI 0.86–7.82)), this difference was not statistically significant (*p* = 0.08). Patients treated with Alteplase had higher odds of an unfavorable mRS (4–5) than the Tenecteplase group upon discharge (Tenecteplase 35.6% vs. Alteplase 37.4%; OR 1.08 (95% CI 0.54–2.18)), this difference is not statistically relevant (*p* = 0.82). The odds of in-hospital mortality (Tenecteplase 15.6% vs. Alteplase 20.1%; OR 1.17 (95% CI 0.49–2.78)) were higher in the Alteplase group rather than in the Tenecteplase group, though this difference was not statistically significant (*p* = 0.73). Patients treated with Tenecteplase had higher odds of excellent functional outcome mRS (0–2) than Alteplase-treated patients after 3 months (Tenecteplase 56.7% vs. Alteplase 39.6%; OR 0.50 (95% CI 0.24–1.06)), though this difference is not statistically significant (*p* = 0.07). Patients treated with Alteplase had higher odds of mRS 3 than the Tenecteplase group after 3 months (Tenecteplase 5.4% vs. Alteplase 32.4%; OR 8.40 (95% CI 1.91–36.88)), and this difference is statistically significant (*p* < 0.001). Patients treated with Alteplase had lesser odds of an unfavorable mRS (4–5) than the Tenecteplase group after 3 months (Tenecteplase 24.3% vs. Alteplase 12.6%; OR 0.45 (95% CI 0.18–1.15)), though this difference is not statistically relevant (*p* = 0.09). The 90-day mortality did not differ (Tenecteplase 13.5% vs. Alteplase 15.3%; OR 1.16 (95% CI 0.40–3.39)), though this difference is not statistically significant (*p* = 0.79). All the statistically significant data previously described in this paragraph can be seen in Table 2. The distribution of the discharge mRS and the 90-day mRS between patients in the Tenecteplase or Alteplase group can be seen in Figure 1 and Figure 2. The OR for the functional outcome (in-hospital mortality, mRS after 90 days) were determined and adjusted for age, sex, pre-stroke mRS, onset-to-EVT-time, and baseline NIHSS—no difference in univariate and multivariate analysis of the binary logistic regression was found in Tenecteplase and Alteplase groups for the adjusted factors listed above in the in-hospital mortality, 90-day mRS 0–2, 3, 4–5 groups. However, in the Alteplase group, a correlation was found between the 90-day mRS 6 group and age, and this difference is statistically relevant (*p* = 0.016). The OR for spontaneous recanalization was determined and adjusted for age, sex, pre-stroke mRS, onset-to-EVT time, and baseline NIHSS—no correlation was found in Tenecteplase and Alteplase groups for the adjusted factors listed above.

## 4. Discussion

Our single-center retrospective observational study showed that intravenous thrombolysis with Tenecteplase (0.25 mg/kg) versus Alteplase (0.9 mg/kg) had better recanalization rates before EVT and better rates of excellent 90-day functional outcome (mRS 0–1) in AIS patients with LVO, who underwent EVT. The in-hospital and 90-day patient mortality did not differ between groups.

The significant increase in the rates of pre-EVT recanalization with Tenecteplase compared to Alteplase is in line with previous studies [6,16]. In addition, some randomized trials reported that final reperfusion rates were significantly better for patients treated with Tenecteplase [3].

Given that EVT is a particular factor in the clinical outcome and 90-day functional independence, pre-EVT recanalization was chosen as the primary outcome to assess the efficacy of the selected intravenous thrombolysis agent. Such a choice was in line with previously mentioned studies.

In addition, successful post-EVT reperfusion (mTICI 2b–3) was achieved in 42 (93.4%) Tenecteplase-treated patients. These findings are slightly better than in the study by Gerschenfeld et al. [17] where final recanalization was achieved in 83.7% of patients using Tenecteplase before EVT. Although post-EVT reperfusion rates did not differ significantly between groups, it should be emphasized that most AIS patients with LVO still require EVT and the data do not suggest the use of Tenecteplase without proceeding to EVT.

Patients who received intravenous Tenecteplase had significantly better excellent 90-day functional outcomes than those who received intravenous Alteplase. Real-world comparison of Tenecteplase and Alteplase highlighted that AIS patients with LVO treated with intravenous Tenecteplase (0.25 mg/kg bolus) had an increased likelihood to achieve neurological improvement but showed no difference in the 90-day functional independence (mRS scores of 0–2) between the two groups [8,17]. Additionally, the rates of 90-day mortality (11% vs. 18%, *p* = 0.703) were similar in the two groups and similar to our data (13.5% vs. 15.3%, *p* = 0.79). Evidence from recently published large nonrandomized studies suggests Tenecteplase is as safe as Alteplase and potentially associated with improved functional outcomes compared with Alteplase [18,19].

In the NOR-TEST study, the 90-day functional outcome (mRS 0–1) was similar between the Tenecteplase and Alteplase groups (49.2% vs. 45.2% in moderate stroke patients and 23.7% vs. 15.6% in severe stroke patients) [20] although a higher dose of Tenecteplase (0.4 mg/kg) was used. However, EXTEND-IA TNK II, NOR-TEST 2A trials did not show non-inferiority with the 0.4 mg/kg Tenecteplase dose. Therefore, Tenecteplase 0.25 mg/kg is likely to be the optimal dose.

In the study by Ole Morten Rønning et al., there were higher rates of the excellent 90-day functional outcome (MRS score of 0–1) with 57% of patients that received Tenecteplase and 53% of patients that received Alteplase [21]. Although it should be taken into consideration that their study included patients with substantially lower median admission NIHSS scores (3 (IQR 2–6).

It should be noted that both groups (Tenecteplase and Alteplase) in our study are comparable in baseline characteristics (e.g., age, sex, comorbidities). Baseline NIHSS (14 vs. 15, *p* = 0.52), pre-stroke mRS, occlusion site (anterior or posterior circulation), and onset time to EVT were without significant differences.

The improvement in the reperfusion rate and the excellent functional outcome did not come with increased complications. A significant intracranial hemorrhage ECASS (PH1, PH2) was observed in seven (15.5%) patients who received Tenecteplase. Randomized controlled clinical trials (NOR-TEST, EXTEND-IA TNK, ATTEST) of Tenecteplase versus Alteplase in AIS patients with LVO have indicated somewhat lower rates of symptomatic ICH, describing a trend toward improved safety profiles [4,6,13,22], but other observational studies have shown even higher rates of symptomatic ICH (21%) [16].

Our study has several limitations. First, as Tenecteplase was not available for routine stroke treatment in 2020, the Alteplase and Tenecteplase groups include patients from two different periods. However, this should not have biased the result, as Tenecteplase is easier to administer, and our CSC has an experience in reperfusion therapies for over 10 years. Second, this is a retrospective non-randomized analysis of prospectively collected data, therefore, selection bias cannot be excluded. Third, since this was a single-center study with rather a small sample size, our findings could have limited generalizability, and some weak statistical associations could have been missed. Fourth, despite adjusting for imbalances in baseline characteristics, there is a risk for residual confounding due to unmeasured confounders. Fifth, the comparison of different historical cohorts, which is prone to selection biases, is also a very important limitation. 

Other things that could be considered in the discussion are that Tenecteplase can be associated with shorter treatment times [5,8], reduced cost in most countries, cost-effective (EXTEND-IA TNK), Tenecteplase thrombolysis is more practical during the pandemic, simpler management during patient transportation to the hospital (MSU trial from ESOC 2022).

## 5. Conclusions

Our real-world single-center retrospective observational study in Latvia supported the fact that Tenecteplase (25 mg/kg) could have had superior clinical efficacy over Alteplase for AIS patients with LVO, administered before EVT. The improvement in the reperfusion rate and better excellent functional outcomes could have come without an increased safety concern.

## Figures and Tables

**Figure 1 medicina-58-01169-f001:**
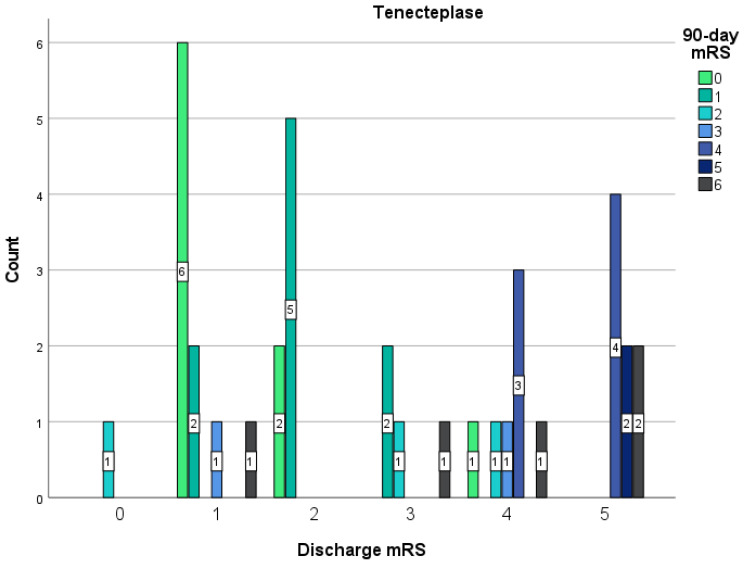
Distribution of the discharge mRS and the 90-day mRS between patients treated with Tenecteplase. Shown is the distribution of scores for disability on the modified Rankin Scale (which ranges from 0 to 6, with higher scores indicating more severe disability, and 6 indicating death) among patients in the Tenecteplase group during discharge and after 90 days. The bars indicate the mRS results after 90-days. The numbers in the bars are the count of patients who had each score after 90-days.

**Figure 2 medicina-58-01169-f002:**
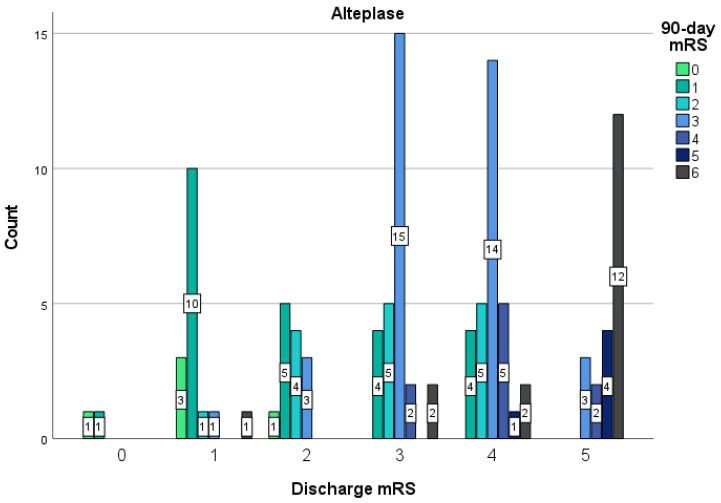
Distribution of the discharge mRS and the 90-day mRS between patients treated with Alteplase. Shown is the distribution of scores for disability on the modified Rankin Scale (which ranges from 0 to 6, with higher scores indicating more severe disability, and 6 indicating death) among patients in the Alteplase group during discharge and after 90 days. The bars indicate the mRS results after 90-days. The numbers in the bars are the count of patients who had each score after 90-days.

**Table 1 medicina-58-01169-t001:** Baseline characteristics.

		Tenecteplase *N* = 45	Alteplase *N* = 139	*p* Value
Age, mean (SD)		68.4 (13.5)	73.0 (11.4)	0.14
Sex, female (%)		24 (53.3)	71 (51.1)	0.79
Hospital stay, median		10	11	0.35
Arterial hypertension, N (%)		39 (86.7)	118 (84.9)	0.77
Atrial fibrillation, N (%)		26 (57.8)	92 (66.2)	0.31
Coronary artery disease, N (%)		11 (24.4)	43 (30.9)	0.41
Congestive heart failure, N (%)		15 (33.3)	71 (51.1)	0.04
Previous cerebrovascular events, N (%)		15 (33.3)	25 (18.0)	0.03
Diabetes mellitus, N (%)		14 (31.1)	31 (22.3)	0.23
Stroke etiology, N (%)	Atherothrombotic	26 (57.8)	98 (70.5)	0.14
	Cardioembolic	12 (26.7)	25 (18.0)	
	Other determined	3 (6.7)	2 (1.4)	
	Undetermined	4 (8.9)	14 (10.1)	
Pre-stroke mRS, median *		0 (0–1)	0 (0–1)	0.26
Baseline NIHSS, median		14 (4–26)	15 (2–31)	0.52
mRS at admission, median **		5	5	0.78
Onset time to thrombectomy, median minutes		165	180	0.32
Thrombectomy, N (%)	Anterior circulation	31 (88.6)	113 (89)	0.95
	Posterior circulation	4 (11.4)	14 (11)	
Pre-EVT recanalization, N (%)		10 (22.2)	12 (8.6)	0.02
Successful reperfusion (mTICI 2b–3) post-EVT, N (%) ***		42 (93.4)	113 (81.3)	0.07
Angioedema N (%)		1 (2.2)	0 (0)	0.08
Hemorrhagic imbibition—ICH, N (%)		15 (33.3)	39 (28.1)	0.50
ICH ECASS score, N (%)	HI1	5 (35.7)	8 (20.5)	0.74
	HI2	2 (14.3)	10 (25.6)	
	PH1	3 (21.4)	13 (33.3)	
	PH2	4 (28.6)	8 (20.5)	
SAH, N (%)		2 (4.4)	9 (6.5)	0.62

Quantitative normally distributed data (Kolmogorov–Smirnov test) are presented as mean (standard deviation (SD)), categorical nominal values are presented with %, frequencies (N, number of patients). *p*-value < 0.05 was considered statistically significant and calculated using a nonparametric Mann–Whitney U Test. SD, standard deviation; IQR, interquartile range; mRS, modified Rankin Scale; NIHSS, National Institutes of Health Stroke Scale; mTICI, modified treatment in cerebral infarction scale; EVT, endovascular treatment, ICH, intracerebral hemorrhage; ECASS, European Cooperative Acute Stroke Study; SAH, subarachnoid hemorrhage. * pre-mRS spread—Tenecteplase: 0 = 86.7%, 1 = 11.1%, 2 = 2.2%; alteplase: 0 = 79.8%, 1 = 10.8%, 2 = 7.9%, 3 = 1.4%. ** mRS at admission spread—Tenecteplase: 3 = 4.4%, 4 = 17.8%, 5 = 77.8%; Alteplase: 3 = 6.5%, 4 = 14.4%, 5 = 79.1%. *** mTICI score after thrombectomy spread—Tenecteplase: 0 = 4.4%, 2a = 2.2%, 2b = 15.6%, 2c = 11.1%, 3 = 66.7%; Alteplase: 0 = 12.9%, 1 = 1.4%, 2a = 4.3%, 2b = 10.1%, 2c = 7.9%, 3 = 63.3%.

**Table 2 medicina-58-01169-t002:** Patient outcomes.

		Tenecteplase *N* = 45	Alteplase *N* = 139	OR (95% CI)	*p* Value
Discharge mRS, N (%)	0–2	18/45 (40)	31/139 (22.3)	0.43 (0.21–0.88)	0.02
90-day mRS, N (%)	3	2/37 (5.4)	36/111 (32.4)	8.40 (1.91–36.88)	<0.001

mRS, modified Rankin Scale; OR, odds ratio; Confidence interval, CI.

## Data Availability

The data presented in this study are available on request from the corresponding author. The data are not publicly available due to data privacy and protection.

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
