# Peer review of "Tenecteplase or Alteplase Better in Patients with Acute Ischemic Stroke Due to Large Vessel Occlusion: A Single Center Observational Study"

_medicina, 2022, doi:10.3390/medicina58091169_

Round 1
Reviewer 1 Report
Dear Authors,
I have had the pleasure of reviewing your article: Tenecteplase or alteplase better in patients with acute ischemic stroke due to large vessel occlusion: a single center study from Latvia
The article is well written. Very good topic. Results encouraging further work. The results and topic of the work are interesting. The work is worth reading, therefore I recommend it for publication.
Author Response
Dear Reviewer,
We thank You very much for the positive report!
Best regards
Research team
Reviewer 2 Report
This is an interesting observational non randomized study on the use of tenecteplase in acute stroke patients with large vessel occlusion from one comprehensive stroke center in Latvia. The study adds to the accumulating evidence from RCTS (see ACT trial DOI: 10.1016/s0140- 6736(22)01054-6.) and real-world data, that are adequately discussed and referred by the authors.
It seems thoroughly well-organized report, however there might be some points this work might benefit of if adjusted:
1. Regarding the efficacy outcomes in terms of consistency with previous trials, consider adding favorable functional outcome (FFO- mRS 0-2) and mRS shift analysis (distribution of the 3-month mRS scores between patients treated with alteplase and tenecteplase was compared by using the Cochran-Mantel Haenszel test).
2. Regarding safety outcomes please provide sICH eg PH2 with clinical deterioration documented by increase >=4 in NIHSS score. Consider adding in the methods section, the imaging protocol (who made the evaluations/was she/he blinded to the outcomes?) and how the hemorrhagic transformation was evaluated (based on CT or MRI?).
3. In order to strengthen the discussion , consider also adding the largest recent metanalysis from real-world data DOI: 10.1001/jamanetworkopen.2022.4506 , and also the largest real-world datasets DOI: 10.1002/ana.26445from the SITS registry and DOI: 10.1177/23969873221113729 from the TETRIS registry in France that were recently published).
4. The main limitation of the study is the comparison of different historical cohorts which is prone to selection biases and this should be acknowledged in the limitations section.
5. There are some typos regarding the references. Consider renumbering.
6. Finally, I suggest rewriting the last paragraph by replacing the bullet points with single text.
Author Response
Dear Reviewer,
We thank You very much for the comments on our manuscript, we considered them and provided the necessary revisions. Please find the point-by-point explanation, and details of our revisions below:
1. Regarding the efficacy outcomes in terms of consistency with previous trials, consider adding favorable functional outcome (FFO- mRS 0-2) and mRS shift analysis (distribution of the 3-month mRS scores between patients treated with alteplase and tenecteplase was compared by using the Cochran-Mantel Haenszel test).
- We adjusted the 90-day mRS division in groups as has been suggested and to be consistent with previous trials. These changes can be seen in the results section of the article.
- mRS shift analysis. We added the discharge mRS score in Table 2. Patient Outcomes for additional information and to be able to evaluate the progress of the outcome more thoroughly. We added the description of the results we got in the results section.
- As suggested, we also did the mRS shift analysis using the discharge mRS and mRS after 3 months in both groups using the Cochran-Mantel Haenszel test and added a graphic explanation of the results below in Table 2 – Figures 1. and 2.
2. Regarding safety outcomes please provide sICH eg PH2 with clinical deterioration documented by increase >=4 in NIHSS score. Consider adding in the methods section, the imaging protocol (who made the evaluations/was she/he blinded to the outcomes?) and how the hemorrhagic transformation was evaluated (based on CT or MRI?).
- Noted, detailed information was added in 2.2. Outcome measures section.
3. In order to strengthen the discussion, consider also adding the largest recent metanalysis from real-world data DOI: 10.1001/jamanetworkopen.2022.4506, and also the largest real-world datasets DOI: 10.1002/ana.26445from the SITS registry and DOI: 10.1177/23969873221113729 from the TETRIS registry in France that was recently published).
- Thank You for the recommendation, noted, and added to the discussion part.
4. The main limitation of the study is the comparison of different historical cohorts which is prone to selection biases, and this should be acknowledged in the limitations section.
- Noted, this limitation has been added to the discussion section.
5. There are some typos regarding the references. Consider renumbering.
- References have been re-checked and renumbered using EndNote 20.
6. I suggest rewriting the last paragraph by replacing the bullet points with single text.
- The last paragraph of the discussion has been rewritten using single text.
We hope we have covered and revised any misunderstandings that were had. Thank You!
Reviewer 3 Report
1. The tiles should include the description of the type of the study. E.g . observational, cross-sectional,…
2. Methods
2.1 Definition of AIS should be provided.
2.2 Could the authors provide the study protocol or register?
2.3 What are the opinions about the area of brain damage (location, size) with the results of the drugs? How was this confounding variable assessed?
2.4 The study was conducted during the coronavirus pandemic. 01/2020 – 12/2021. The number related to COVID-19 ischemic events should be reported.
2.5 Who made the diagnosis of AIS? Who performed alteplase and tecneteplase?
3. Statistics
3.1 How the authors calculated the power of the study.
3.2 How was the data distributed?
3.3 How were confounding variables assessed?
3.4 Edition of IBM SPSS should be provided.
4. results
4.1 The reviewer was not able to find the results of the logistic regression. It is advised to provide significant results in a table and specify the model and variables used.
4.2 The reviewer believes that the authors did a chi-square in the table about the baseline characteristics. The number of chi-squares appears to be higher, correlated with group distinction. Could the authors explain more about it? Because there is a natural difference between the groups. Could this not explain the results encountered?
4.3 In the discussion, it is advised to provide an entire paragraph. The bullets should be removed.
5. The Grammatical English should be revised. It is advised professional editing service.
6. I would advise including a table comparing other studies in the literature about this theme.
7. The conclusion should be rephrased. The present format could lead to misunderstanding.
The idea of the manuscript is interesting. But, the authors should improve grammar and statistics. It is advised the assistance of a person with statistical background to revise the manuscript.
Author Response
Dear Reviewer,
We thank You very much for the comments on our manuscript, we considered them and provided the necessary revisions. Please find the point-by-point explanation, and details of our revisions below:
- The tiles should include the description of the type of the study. E.g. observational, cross-sectional,…
- The title has been corrected.
- Definition of AIS should be provided.
- The definition of acute ischemic stroke (AIS) has been provided in the Introduction; 1st
- Could the authors provide the study protocol or register?
- We have added additional information on the study protocol in 2.2. Outcome measures section.
- What are the opinions about the area of brain damage (location, size) with the results of the drugs? How was this confounding variable assessed?
- This was not one of our study endpoints, therefore it has not been mentioned.
- The study was conducted during the coronavirus pandemic. 01/2020 – 12/2021. The number related to COVID-19 ischemic events should be reported.
- Covid-19 positive patients were not included in the study. For no further confusion, we have added this statement in 2.1. Study design section.
- Who made the diagnosis of AIS? performed alteplase and Tenecteplase?
- The patients with AIS were diagnosed by a Stroke Neurologist, who further on performed a thrombolysis procedure. This information was added in 2.2. Outcome measures section. We also added additional information that thrombectomies were performed by interventional radiologists in the same section, so it is clearer.
Statistics: We did a repeated statistical evaluation of the data and had a consultation with the Riga Stradiņš University Biostatistics laboratory expert on statistics. 2.3. Statistical analysis section has been edited for better understanding.
- How the authors calculated the power of the study.
- Since we used nonparametric tests to evaluate the variables, there is no analytic function, that can calculate the power, therefore it was unnecessary to be done. While nonparametric tests typically have the same type I error across a very wide class of distributions (in many cases, for all continuous distributions), they don't have the same power characteristics across all distributions. So, while the basic idea is the same -- we specify a particular alternative at which we want a particular amount of power and then we work out the sample size that will give us that rejection rate for that alternative -- to be able to compute the power, we need to specify precisely what the situation is. If we do specify it precisely, then we don't necessarily have to be able to do the calculation algebraically (though sometimes it can be doable); simulation is generally sufficient.
- How was the data distributed?
- Continuous variables (eg. age, hospital stay, onset time to thrombectomy) did not follow the normal distribution, therefore nonparametric tests were used moving forward.
- How were confounding variables assessed?
- We assessed them using the bivariate logistic regression.
- Edition of IBM SPSS should be provided.
- Noted, corrected.
- The reviewer was not able to find the results of the logistic regression. It is advised to provide significant results in a table and specify the model and variables used.
- The results of the logistic regression can be seen in Table 2. Patient Outcomes. The results and variables used were specified in the Result section:
The OR for the functional outcome (in-hospital mortality, mRS after 90 days) were determined and adjusted for age, sex, pre-stroke mRS, onset-to-EVT-time, and baseline NIHSS – no difference in univariate and multivariate analysis of the binary logistic regression was found in Tenecteplase and alteplase groups for the adjusted factors listed above in the in-hospital mortality, 90-day mRS 0–2, 3, 4–5 groups.
- The reviewer believes that the authors did a chi-square in the table about the baseline characteristics. The number of chi-squares appears to be higher, correlated with group distinction. Could the authors explain more about it? Because there is a natural difference between the groups. Could this not explain the results encountered?
- Two baseline characteristics have a statistically significant difference between the two groups. First is congestive heart failure (CHF), which shows that in the Alteplase group there is a higher chance of the patients having CHF. Considering the fact, that the mean age in both groups is roughly around 70, the likelihood of having other comorbidities at this age, such as CHF, is high. We ought to think that the fact that this number is higher in the Alteplase group is just a coincidence and gives us no reason to think that it could encounter the results moving forward. The other characteristic is previous cerebrovascular events (such as stroke, and transient ischemic attacks), which shows that in the Tenecteplase group there is a higher chance of having previous cerebrovascular events. Considering the fact, that the Tenecteplase group showed improvement in reperfusion rate and better excellent functional outcomes, it gives us no reason to think that this difference had an impact on the results.
- In the discussion, it is advised to provide an entire paragraph. The bullets should be removed.
- Noted, corrected.
- The Grammatical English should be revised. It is advised professional editing service.
- Noted, the grammar has been checked and edited by professional editing service.
- I would advise including a table comparing other studies in the literature about this theme.
- We are not performing a literature review; therefore, this type of table is unnecessary in our opinion.
- The conclusion should be rephrased. The present format could lead to misunderstanding.
- Noted, corrected.
We hope we have covered and revised any misunderstandings that were had. Thank You!
Round 2
Reviewer 3 Report
I congratulate the authors on the revised version of the manuscript. But, there are some concerns regarding grammatical English, methods, and statistical analysis.
1. There are some concerns regarding grammatical English. It is advised professional editing service. If the authors have already sent for editing service, they can include the certificate for review as non-published supplementary material.
For E.g. long phrases could lead to misunderstanding. In the abstract, the description of the methods should be rephrased. It is difficult to understand that the authors are using mTICI.
2. It is advised to include the full description of every term at the first presentation. E.g. NIHSS, TICI, EVT, LVO.
3. The format of the present manuscript should be revised. There are no tab spaces at the beginning of the paragraph.
4. Methodology and statistical analysis
It is advised expert opinion of statistics to describe methodology and statistics. If the authors already did, they should upload the certificate of statistical analysis as non-published supplementary material.
In the reviewer’s opinion, this manuscript type is better described as a “Cross-sectional, Retrospective Observational Study.”
Please specify inclusion and exclusion criteria.
It is advised to specify variables collected in the methods. If a questionnaire was used, the authors should upload it as published supplementary material.
The number of individuals assessed should be included in the study should be described in the methods.
The authors should include the locations where the study occurred.
Statistics
The information discussed in the first round should be included in the manuscript.
Was data normally or non-normally distributed? This defines the best statistical method to use. If it was non-normal distributed, authors should upload the graphs rendered by the statistical program as published supplementary material for further analysis.
A study supporting the division of 90-day mRS scores into groups should be cited.
What were the variables assessed in the logistic regression? Did the authors use any model for the regression?
It is advised to revise the power of the study. Non-parametric studies lead to a significant decrease in power. The authors should provide some evidence of what they expected to encounter.
For further understanding of the bivariate regression, Table 2 should only include significant results.
Could the authors publish their spreadsheet database in the Mendeley data?
https://data.mendeley.com/
For further improvement of the manuscript, it is advised for the authors to upload as non-supplementary material STROBE guideline checklist regarding observational studies.
https://www.equator-network.org/reporting-guidelines/strobe/
Author Response
Dear Reviewer,
We thank You again for the comments on our manuscript, we considered them and provided the necessary revisions. Please find the point-by-point explanation and details of our revisions below:
- There are some concerns regarding grammatical English. It is advised professional editing service. If the authors have already sent for editing service, they can include the certificate for review as non-published supplementary material.
For E.g. long phrases could lead to misunderstanding. In the abstract, the description of the methods should be rephrased. It is difficult to understand that the authors are using mTICI.
- Unfortunately, the editing service cannot provide a certificate.
- We are using the modified treatment in cerebral infarction (mTICI) scale. For better understanding we have made necessary corrections.
- It is advised to include the full description of every term at the first presentation. E.g. NIHSS, TICI, EVT, LVO.
- The first presentation of the terms can be seen in the abstract, where we do not provide a full description of all terms because it is not usually done in the abstract section. As for the actual manuscript – noted, corrections have been made.
- The format of the present manuscript should be revised. There are no tab spaces at the beginning of the paragraph.
- Noted, corrected.
- It is advised expert opinion of statistics to describe methodology and statistics. If the authors already did, they should upload the certificate of statistical analysis as non-published supplementary material.
- We consulted various experts in the statistical field from The Statistics Unit of Riga Stradins University Faculty of Medicine. The first expert that provided their services was Jeļena Larina, Scientific Consultant, Statistics Unit. The second was Acting Lecturer, Statistician, Statistics Unit, Eva Petrošina. Both consultants are experts in their field with the required education and experience with Scientific work. Unfortunately, we cannot provide a certificate of statistical analysis as our Statistical Unit does not provide one. In addition, all authors have taken necessary courses in Statistics as a part of their education.
- Considering all that, we have made minor changes to the methodology and statistics section.
- In the reviewer’s opinion, this manuscript type is better described as a “Cross-sectional, Retrospective Observational Study.”
- Noted, corrected.
- Please specify inclusion and exclusion criteria.
- Noted, added in the 2.1. Study design and population paragraph. We based our evaluation on inclusion criteria, exclusion criteria revolve solely around inclusion criteria, therefore we do not think it is necessary to be explained.
- It is advised to specify variables collected in the methods. If a questionnaire was used, the authors should upload it as published supplementary material.
- A questionnaire was not used; therefore, no supplementary material will be added. We have specified variables collected in the Methods section.
- The number of individuals assessed should be included in the study should be described in the methods.
- Noted, added in the 2.1. Study design and population paragraph.
- The authors should include the locations where the study occurred.
- Noted, added in the 2.1. Study design and population paragraph.
- The information discussed in the first round should be included in the manuscript.
- We have made the appropriate corrections and incorporated them into the manuscript.
- Was data normally or non-normally distributed? This defines the best statistical method to use. If it was non-normal distributed, authors should upload the graphs rendered by the statistical program as published supplementary material for further analysis.
- We have answered this question in Round one already - How was the data distributed?
- Continuous variables (eg. age, hospital stay, onset time to thrombectomy) did not follow the normal distribution, therefore nonparametric tests were used moving forward.
- The distribution analysis is done using basic statistical methods and the best statistical method is further on evaluated by the authors, as well as checked by our Statistical experts. We do not think there is a need for publishing such graphs, as well as there is no such practice in other similar high-scale studies. Considering the above, we have made changes to the statistics section for a better understanding of the methods and tests we used. We have explained all the necessary information in the Statistics paragraph.
- A study supporting the division of 90-day mRS scores into groups should be cited.
- Good outcome following stroke is commonly defined as scores 0 – 2 and poor outcome scores of 3 – 6 (although there is a big difference between groups 3 and 6 – hence our division). There is no study supporting the exact division of the mRS groups. All studies usually divide each mRS group separately, though one common division group seen throughout studies is mRS score 0-2 – functional independence. The division of 0-2 / 3 / 4-5 / 6 in our opinion best highlights the functional outcome.
- What were the variables assessed in the logistic regression? Did the authors use any model for the regression?
- This question was also answered in Round one – The reviewer was not able to find the results of the logistic regression. It is advised to provide significant results in a table and specify the model and variables used.
- The results of the logistic regression can be seen in Table 2. Patient Outcomes. The results and variables used were specified in the Result section: The OR for the functional outcome (in-hospital mortality, mRS after 90 days) were determined and adjusted for age, sex, pre-stroke mRS, onset-to-EVT-time, and baseline NIHSS – no difference in univariate and multivariate analysis of the binary logistic regression was found in Tenecteplase and alteplase groups for the adjusted factors listed above in the in-hospital mortality, 90-day mRS 0–2, 3, 4–5 groups.
- This information can be found in our manuscript in the Result section, but we have added this information in the Statistics section for better understanding.
- All efficacy and safety outcomes of interest were further assessed in multivariable binary and ordinal multivariable logistic regression models adjusting for the confounders - age, sex, pre- mRS, onset-to-EVT-time, and baseline NIHSS.
- It is advised to revise the power of the study. Non-parametric studies lead to a significant decrease in power. The authors should provide some evidence of what they expected to encounter.
- No other study that we can find that uses non-parametric tests for non-normally distributed data has described the power of the study or why they have done it. It is a basic statistical principle that there is no analytic function that can correctly calculate the power in these cases, therefore it is unnecessary to be done or described in our opinion. As we have mentioned, we use non-parametric tests, and it follows that the power has not been evaluated and the study has decrease in power.
- For further understanding of the bivariate regression, Table 2 should only include significant results.
- Noted, corrections have been made to Table 2.
- Could the authors publish their spreadsheet database in the Mendeley data? https://data.mendeley.com/.
- Unfortunately, no, we do not think it is necessary to provide our data spreadsheet as all our results have been explicitly described and shown in the result section.
We hope we have covered and revised any misunderstandings that were had. Thank You!
Round 3
Reviewer 3 Report
The reviewer would like to congratulate the authors for the improvement throughout the reviews. The authors answered all the reviewer's queries.